# Extracellular Matrix and Fibrocyte Accumulation in BALB/c Mouse Lung upon Transient Overexpression of Oncostatin M

**DOI:** 10.3390/cells8020126

**Published:** 2019-02-05

**Authors:** Fernando M. Botelho, Rebecca Rodrigues, Jessica Guerette, Steven Wong, Dominik K. Fritz, Carl D. Richards

**Affiliations:** McMaster Immunology Research Centre, Department of Pathology and Molecular Medicine, McMaster University, Hamilton, ON L8S 4L8, Canada; botelhf@mcmaster.ca (F.M.B.); rebeccamrodrigues@gmail.com (R.R.); jessica.guerete@gmail.com (J.G.); swong2018@meds.uwo.ca (S.W.); fritz.dkp@gmail.com (D.K.F.)

**Keywords:** inflammation, fibrocytes, ECM accumulation, cytokines, Oncostatin M, fibrosis

## Abstract

The accumulation of extracellular matrix in lung diseases involves numerous factors, including cytokines and chemokines that participate in cell activation in lung tissues and the circulation of fibrocytes that contribute to local fibrotic responses. The transient overexpression of the gp130 cytokine Oncostatin M can induce extracellular matrix (ECM) accumulation in mouse lungs, and here, we assess a role for IL-13 in this activity using gene deficient mice. The endotracheal administration of an adenovirus vector encoding Oncostatin M (AdOSM) caused increases in parenchymal lung collagen accumulation, neutrophil numbers, and CXCL1/KC chemokine elevation in bronchioalveolar lavage fluids. These effects were similar in IL-13-/- mice at day 7; however, the ECM matrix induced by Oncostatin M (OSM) was reduced at day 14 in the IL-13-/- mice. CD45+col1+ fibrocyte numbers were elevated at day 7 due to AdOSM whereas macrophages were not. Day 14 levels of CD45+col1+ fibrocytes were maintained in the wildtype mice treated with AdOSM but were reduced in IL-13-/- mice. The expression of the fibrocyte chemotactic factor CXCL12/SDF-1 was suppressed marginally by AdOSM in vivo and significantly in vitro in mouse lung fibroblast cell cultures. Thus, Oncostatin M can stimulate inflammation in an IL-13-independent manner in BALB/c lungs; however, the ECM remodeling and fibrocyte accumulation is reduced in IL-13 deficiency.

## 1. Introduction

Chronic lung inflammatory diseases such as asthma, chronic obstructive pulmonary disease (COPD), and pulmonary fibrosis collectively affect a significant proportion of patients in North America. At various stages of these conditions, the pathological excess of extracellular matrix (ECM) can compromise lung function, and thus, the effects of key molecules and cytokines including TGF-β on ECM deposition has been keenly investigated [1,2,3]. TGF-β clearly acts in milieu of other factors in vivo, and although thought to be a central mediator of ECM accumulation and fibrotic mechanisms including epithelial to mesenchymal transition (EMT) [3,4], other cytokines may be involved. As suggested by numerous studies (reviewed in Reference [5]), members of the gp130 cytokine family including IL-6, IL-11, and Oncostatin M (OSM) appear to participate in the mechanisms involved in lung ECM remodeling [6,7,8]. The OSM-induced matrix accumulation in lung was shown to be independent of the TGF-beta signaling in mouse models [8]. Gp130 cytokines robustly activate STAT3 cell signaling, and in genetic models in mice, the over-activation of STAT3 renders animals much more sensitive to the ECM remodeling effects in the bleomycin model of lung fibrosis [9]. Furthermore, this model is independent of the canonical TGFb signaling pathway SMAD3, emphasizing other pathways can lead to lung fibrosis in vivo [9].

As a member of the gp130 cytokine family, OSM is a multifunctional cytokine that can regulate homeostatic functions as well as disease processes [10,11,12]. Mouse OSM engages in receptor complexes that include the gp130 signaling molecule and the OSMRβ chain [13,14,15], both of which are broadly expressed in connective tissue cells. OSM has been shown to induce a number of responses that regulate the remodeling of ECM in articular joints [16,17,18], skin [19,20], and bones [21,22]. OSM induces the ECM remodelling of lungs in animal models [8,23,24] and may contribute significantly to chronic inflammatory lung disease in humans since it is found at elevated levels locally in sputum samples from severe asthmatics [25] and in the bronchoalveolar lavage fluid (BALF) of idiopathic pulmonary fibrosis patients [8]. OSM can also regulate chemokine release from cells derived from the lung, including fibroblasts [26], airway smooth muscle cells [27], as well as airway epithelial cells [28,29,30].

There is increasing evidence that conditions that result in ECM remodeling involve the activation of a number of cell types, including resident fibroblasts, myofibroblasts, and circulating fibrocytes [31,32,33]. Chemokines such as CXCL12/SDF-1 have chemotactic activities for fibrocytes [34,35] and may play a role in generating the fibrocyte response and local fibrogenic sequale. It is currently unclear whether OSM induces chemokine production and subsequent fibrocyte accumulation for these cells.

Th2 cytokines have been implicated in inducing the ECM accumulation in lungs [36]. The overexpression of IL-13 using the adenovirus vector for pulmonary transgene expression induced marked increases in the inflammatory ECM deposition [37]. We have shown that the transient pulmonary transgene expression of OSM induces lung ECM in C57Bl/6 mice [23], although the mechanisms may be quite different in BALB/c mice [24]. In other models of airway inflammation, BALB/c mice have been typically characterized as more biased toward Th2 immune responses. Thus, we have examined here the BALB/c responses to the transient pulmonary overexpression of OSM, their dependence on IL-13, and the accumulation of fibrocytes and CXCL12/SDF-1 in lung tissue.

## 2. Materials and Methods

### 2.1. Animals

Female BALB/c mice (8–10 weeks old) were purchased from Charles River Laboratory (Ottawa, ON, Canada). Female BALB/c IL-13-/- mice were a courtesy of Dr. Waliul Khan (McMaster University). All mice were housed in standard conditions with food and water ad libitum. All procedures were approved by the Animals Research Ethics Board at McMaster.

### 2.2. Administration of Adenovirus Constructs and Sample Collection

Wildtype and IL-13-/- mice were administered 5 × 10^7^ pfu of replication deficient AdDl70 or Ad encoding OSM (AdOSM) through the endotracheal route of administration as previously published [23,38]. Mice were euthanized after 5, 7, or 14 days and bled, and alveolar lavage was performed as previously described [23,38]. Alveolar lavage was centrifuged, and supernatants were stored for future analysis by ELISA. Cell pellets were resuspended, counted, and subjected to cytocentrifugation at 300 rpm for two minutes. Differential counts were determined after staining with protocol Hema3 (Fisher Scientific, Ottawa, ON, Canada). Left lungs were perfused with 10% formalin and fixed for 48 h subsequent to histological preparation and histochemical staining. Right lungs were snap frozen and stored at −80 °C for RNA extraction.

### 2.3. Reagents and Cell Culture

Mouse lung fibroblast (MLF) cultures were generated from BALB/c or C57Bl/6 mouse lungs as previously described [26]. LA-4 and A549 cells were purchased from ATCC and cultured under recommended conditions. Recombinant E. coli derived cytokines (mouse OSM, IL-1β, and IL-4) were purchased from R&D systems.

### 2.4. ELISA and Extracellular Matrix Analysis

Duoset ELISA kits were purchased from R&D Systems (Minneapolis, MN, USA) to measure the protein levels of CXCL1/KC, mouse IL-6, mouse CXCL12/SDF-1, mouse CCL11/Eotaxin-1, and mouse VEGF in BALF samples stored at −20 °C. The quantification was completed as per the manufacturer’s instructions. For the measurement of collagen accumulation, the intensity of picrosirius-stained tissue sections under polarized light was quantified from images (>20 images per lung) by using the Ashcroft Method [39] as previously described for BALB/c mouse lungs by Wong et al. [24].

### 2.5. RNA Extraction

Lung tissues or cell culture extracts were homogenized in Trizol (Invitrogen, Life Technologies). RNA was reverse transcribed, and the levels of CXCL12/SDF-1 and IL-6 were assessed by real time Q-PCR (Taqman) using primers with FAM-5′ end-labeled fluorogenic probes for CXCL12/SDF-1 and IL-6 and VIC-5′ end-labeled fluorogenic probes for 18S. mRNA expression was expressed as the ΔΔCt (cycle threshold) values for CXCL12/SDF-1 or IL-6 relative to that of the 18S control RNA. All obtained gene expression assays came from Applied Biosystems, Thermos-Fisher and were performed as previously described [40].

### 2.6. Isolation of Lung Mononuclear Cells and Flow Cytometric Analysis

Lung mononuclear cell suspensions were generated by mechanical mincing and collagenase digestion. Debris were removed by passage through a 40 micrometer screen size nylon mesh (this results in primarily CD45+ hematopoietic cells introduced to the flow cytometry analysis), and cells were resuspended in phosphate-buffered saline (PBS) containing 0.3% bovine serum albumin (BSA) (Invitrogen, Burlington, ON, Canada) or in RPMI media supplemented with 10% fetal bovine serum (FBS) (Sigma-Aldrich, Oakville, ON, Canada), 1% L-glutamine, and 1% penicillin/streptomycin (Invitrogen, Burlington, ON, Canada). Washed once with PBS/0.3% BSA and stained with primary antibodies directly conjugated to fluorochromes for 30 min at 4 °C were 1 × 10^6^ lung mononuclear cells. Acquired on an LSR II (BD Biosciences, San Jose, CA, USA) flow cytometer were 10^5^ live events, and the data were analyzed with FlowJo analysis software (FlowJo, LLC., Ashland, OR, USA). Side scatter and forward scatter parameters were used to define live cell and lymphocyte gates. All antibodies were purchased from BD Biosciences (San Jose, CA, USA) or eBiosciences (San Diego, CA, USA) unless otherwise stated. The following antibodies were used for flow cytometric analysis: APC-cy7-conjugated anti-CD45, PerCP-cy5.5-conjugated anti-CD11c, PE-conjugated anti-CD11b, PE-cy7-conjugated anti-DX5, Pacific Blue-conjugated anti-CD3, and Pacific Orange-conjugated anti-Gr-1. Neutrophils were defined as CD45+, CD11b-hi, and Gr-1+. Macrophage cells were defined as being CD11b+, CD11c-, DX5-, CD3-, and Gr-1-. For the fibrocyte analysis, CD45+ Collagen I (Col1+) fibrocytes were detected first by surface staining the cells with APC-cy7-conjugated anti-CD45 for 30 min at 4 °C and followed by intracellular staining with rabbit anti-collagen I (Rockland, Gilbertsville, PA, USA) and then FITC-conjugated anti-rabbit (Jackson ImmunoResearch, Westgrove, PA, USA) antibodies, each for 30 min at 4 °C in 1× Perm/Wash buffer (BD Biosciences, San Jose, CA, USA) with washes in 1x Perm/Wash between intracellular staining steps. Cells were then washed with 1× PBS/0.3% BSA prior to analysis on an LSR II flow cytometer.

### 2.7. Statistics

Data were analyzed using GraphPad Prism version 5.1 software and presented as mean +/− standard error of the mean (SEM). For in vivo experiments, five animals per group were utilized. One-way analysis of variance (ANOVA) was used to determine the statistical significance, which was defined as *p* < 0.05 using GraphPad Prism. The p-values are indicated in the individual figures.

## 3. Results

We have previously shown that intranasal or endotracheal administration of the adenovirus vector expressing mouse OSM in C57Bl/6 and in BALB/c mice causes the transient pulmonary gene transduction of OSM, which leads to a pronounced remodeling of the lung, including relatively rapid increases in both the collagen gene expression and protein [24]. The AdOSM vector induces OSM levels in BALF transiently (250 +/− 25 pg/mL at day 2 and 1540 +/− 150 pg/mL at day 7) whereas levels in naïve or AdDl70 animals were undetectable. In Figure 1A, we examined the histopathology of wild type BALB/c and IL-13-/- BALB/c mice 7 days after delivery of AdOSM. AdOSM-treated mice showed a disruption of the lung architecture, with thicker alveolar walls compared to the AdDl70-treated counterparts. Representative images of the picrosirus red-stained histological sections are shown and indicate a qualitative increase in the staining within the lung parenchyma in AdOSM-treated mice from both the wildtype and IL-13-/- strains. The quantification of PSR-stained sections showed similar increases in the picrosirius red-staining in the parenchyma of both the AdOSM-treated wildtype and IL-13-/- mice (Figure 1A, right panel). We further assessed the histopathology and parenchymal collagen accumulation after a prolonged 14-day time point with AdOSM in the BALB/c wildtype and IL-13-deficient mice. As shown in Figure 1B, the thickening of the lung architecture and the collagen accumulation in AdOSM-treated mice were significantly reduced in IL-13-/- mice as compared to the wildtype mice. Ashcroft scores of the PSR-stained sections also showed a significant reduction in the collagen staining in IL-13-/- mice as compared to the wildtype mice.

To determine if the inflammation induced by OSM was affected by the absence of IL-13, the BALF fluid was collected, cell frees supernatants were stored, and cytocentrifuged cells were assessed by differential staining. Figure 2A shows that macrophage and neutrophil numbers and percentages in BALF were significantly elevated in the AdOSM-treated BALB/c mice compared to the control vector AdDl70-treated mice.

Similar levels of macrophages and neutrophils were observed in the IL-13-/- in response to the AdOSM treatment in comparison to the wildtype mice. Similar to previous works comparing responses to AdOSM in BALB/c and C57Bl/6 mice [23,24,41], little to no eosinophils were detected in the BALF in either the wildtype BALB/c or IL-13-/- mice in response to AdOSM administration. The chemokine CXCL1/KC (chemoattractant for neutrophils) levels were induced by AdOSM with no difference in the IL-13-/- mice, and the CCL11/Eotaxin-1 levels were low/undetectable and unaltered between the different treatment groups. The vascular endothelial growth factor (VEGF) was also increased in AdOSM-treated mice similarly in both wildtype and IL-13-/- mice (Figure 2B).

To examine the accumulation of CD45 positive cells in this system, flow cytometry was used to assess the different populations of the whole lung upon collagenase digestion and the generation of single cell preparations. Using the strategy shown in Figure 3A, we were able to detect CD45+ collagen1A1+ (CD45+ coll+) cells in the BALB/c lung.

The numbers of these cells were quantified in Figure 3B. The results indicate that similar numbers of CD45+ coll1+ cells were evident in both the uninfected naïve and control AdDl70-treated mice and elevated in cell numbers upon AdOSM administration at day 7. The numbers of CD45+coll1+ cells were sustained at day 14 in the wildtype mice and decreased to basal levels in the IL-13-/- mice. As a comparator, tissue macrophage levels (defined as CD11b+, CD11c-, Gr-1-, DX5-, and CD45+) were similar at day 7 or day 14 between the wildtype and IL-13-deficient mice (Figure 3B). We have previously observed significant increases in alternatively activated/M2 macrophage markers in C57Bl/6 mice treated with AdOSM [40]. However, as shown in Appendix A, Arginase-1 mRNA was not significantly elevated in AdOSM-treated BALB/c lungs, while C57Bl/6 mice showed marked increases (*p* < 0.005), consistent with previous results [40]. In addition, the mRNA from alveolar macrophages retrieved from AdOSM-treated BALB/c mice did not show any difference in Arginase-1 or CD206 mRNA (both markers of M2 macrophage phenotypes) from those alveolar macrophages retrieved from AdDl70-treated control animals.

We then assessed the expression of CXCL12/SDF-1 (chemotactic for fibrocytes) and IL-6 (an inflammatory cytokine, previously shown to be induced by OSM in C57Bl/6 mice [38]). In BALF, as assessed by ELISA, CXCL12/SDF-1 showed a trend of decreased levels in animals treated with AdOSM in either the wildtype or IL-13-/- mice (Figure 4A).

In contrast, the levels of IL-6 in BALF were elevated in AdOSM-treated mice in both the wildtype and IL-13-/- mice, relative to the AdDl70 controls. Assessing the mRNA expression (Figure 4B) in lung tissue at day 7 following the AdDl70- or AdOSM-treatments of the wildtype show that the levels of CXCL12/SDF-1 mRNA were not increased in response to OSM, while the IL-6 mRNA expression levels were elevated in AdOSM-treated mice.

To assess the apparent downregulation of CXCL12/SDF-1 by OSM in an in vitro system, MLF cultures from BALB/c mice were isolated and stimulated with varying concentrations of recombinant (E. coli derived) OSM (Figure 5A, left panel).

Levels above 300 pg/mL of the CXCL12/SDF-1 protein were detected in the unstimulated cells; however, the levels decreased in cells stimulated with 2.5 ng/mL of OSM or higher. The same supernatants contained increased IL-6 levels, stimulated by OSM in a dose-dependent manner. To assess whether the mammalian-expressed mouse OSM had the same effect, MLF cultures were stimulated with the supernatant from A549 cells infected with either the AdDl70 control or AdOSM vector. Figure 5A (right panel) shows the levels of both thee CXCL12/SDF-1 and IL-6 in MLF supernatants after 24 h of stimulation with various dilutions of the Ad-infected A549 supernatants. Both the CXCL12/SDF-1 and IL-6 levels were unregulated in the MLF cultures treated with Addel-70-infected supernatants, whereas the AdOSM-infected supernatants caused a decrease in the CXCL12/SDF-1 levels and a marked increase in the IL-6 levels in a dose-dependent fashion in the MLF cultures. Thus, both the E. coli-derived OSM and mammalian cell-expressed OSM were able to potently suppress the SDF1 production in MLF cultures.

To determine if the CXCL12/SDF-1 protein decrease in vitro in BALB/c mice was also observed in other mouse strains, the total lung RNA and supernatants were assessed by q-RT-PCR and ELISA, respectively in C57Bl/6-derived MLF cells. OSM induced a reduction in the CXCL12/SDF-1 basal levels of proteins in the supernatants (Figure 5B) and CXCL12/SDF-1 mRNA expression (Figure 5C, left panel) following a 24-h treatment of C57Bl/6 MLF cells. In contrast, we observed elevated levels of IL-6 mRNA expression in these cells (Figure 5C, right panel).

## 4. Discussion

The regulation of ECM remodeling in lungs involves the participation of various cytokines and cells. In the model system explored here, we have shown that the Ad-vector-mediated overexpression of OSM in BALB/c mice induces neutrophil infiltration and CXCL1/KC chemokine levels into bronchoalveolar spaces in a manner independent of IL-13 at day 7. We observed that AdOSM induces the accumulation of parenchymal collagen that was evident at day 7 (consistent with previous work [24]) and was further elevated at day 14. The collagen increase was evident in IL-13-/- mice at day 7 but was markedly reduced at day 14. We also observed the fibrocyte accumulation at day 14 as defined by CD45+Coll1+ cells in the lungs, which was also reduced in the IL-13-/- mice. The CXCL12/SDF-1 chemotactic factor for fibrocytes was suppressed by AdOSM in vivo and in OSM-stimulated mouse lung fibroblasts in vitro.

The lack of requirement for IL-13 during the early matrix deposition observed here (day 7) is consistent with the studies done on C57Bl/6 mice, where Mozafarrian et al. showed that OSM induced ECM remodeling in the lungs of similar mice that were also treated with IL-13R.Fc (blocks only IL-13 and not IL-4) or the control Fc administered intraperitoneally [8]. IL-13R.Fc did block the eosinophil accumulation. Such results in C57Bl/6 mice were also consistent with subsequent data showing that the AdOSM-induced elevation of ECM was independent of STAT6, a major signaling pathway of both IL-4 and IL-13 but was required for the effects on eosinophil accumulation [23]. However, as we have previously published [24], the AdOSM vector induces significant parenchymal collagen in BALB/c mice but not a detectable increase in the BALF levels of IL-4/IL-13, CCL24/Eotaxin-2, or the eosinophil accumulation. Collectively, the data suggests that eosinophils are not involved in the ECM increases in either BALB/c or C57Bl6 lungs due to OSM overexpression. We have examined the expression of macrophage markers (by mRNA analysis) in BALB/c mouse lungs treated with AdOSM and have not observed any significant changes in alternatively activated “M2” markers (see Appendix A). In contrast, C57Bl/6 mice show elevated CD206+ cells and arginase-1 expression (as typical M2 macrophage products) in response to AdOSM [40]. Other macrophage phenotypes may be involved in BALB/c lungs in this system; however, this would require further investigation.

AdOSM expression increased the CXCL1/KC levels and neutrophils numbers in BALF, and the lack of difference in the IL-13-/- mice indicated that the ECM effects and neutrophil presence correlate in the BALB/c system here. There is a possibility that the sources of endogenous OSM such as that from neutrophils as well as macrophages contribute to the load of OSM protein in lung inflammatory conditions. Although activated macrophages and T cells are major sources of OSM, Grenier et al. [42] have found that neutrophils can synthesize OSM and release preformed OSM, and that neutrophils from patients with acute lung injury release OSM. Thus, the effects in this model system in BALB/c mice may reflect the OSM generated by the adenovirus vector and endogenous OSM from inflammatory cells.

Fibrocytes have been implicated in fibrotic mechanisms in both mouse and human systems [32,35,43]. These cells have been shown to be recruited to mouse lungs in models of bleomycin-induced pulmonary fibrosis through CXCL12/SDF-1 [35], which interacts with CXCR-4. IL-4 and IL-13 have been shown to stimulate the fibrocyte production of matrix components [44]. Our results indicate that OSM can engage the recruitment of CD45+col1+ fibrocytes to the lung and may contribute to the remodeling of ECM in BALB/c lungs. The correlation between the reduction of fibrocytes at day 14 in the IL-13-/- mice and the reduction in parenchymal collagen is consistent with this. The accumulation of fibrocytes over the time course of the ECM accumulation in this model may be affected at other time points in IL-13KO mice. The association with the decreased fibrocyte number and decreased ECM at day 14 does not show causality, and further experimentation to determine if fibrocytes are critical to the ECM accumulation at day 14 would be required. The role of IL-13 in the fibrocyte accumulation is not clear. Since we cannot detect the induction of IL-13 protein (BALF ELISA, data not shown), it may be that the required IL-13 concentration locally in the lung is low. It is possible that IL-13 (or CXCL12/SDF-1) is elevated at specific times not captured by the present time-points analysed or indeed in specific populations (such as separate macrophage phenotypes). Assessing specific macrophage populations or other cells over the time course of the model development would assist with this and would be the subject of future experimentation. Alternatively, IL-13 may be required at other organ sites to maximally support fibrocyte generation or recruitment. In addition, IL-13 may act directly on the local matrix synthesizing cells, such as fibroblasts, where low levels may enable maximal OSM regulation of collagen synthesis.

Previous work has shown that human OSM could induce the CXCL12/SDF-1 protein expression in vitro in cardiac myocytes and cardiac fibroblasts [45]. In our system using AdOSM in vivo, we did not observe increases in CXCL12/SDF-1. Furthermore, we observed a suppression of CXCL12/SDF-1 by OSM in the in vitro analysis of the MLF responses. Our contrasting results may be due to the cell type/tissue source studied or to possible differences between species responses. There is a possibility that since the ELISA is specific for CXCL12/SDF-1α and not CXCL12/SDF-1β, we are missing CXCL12/SDF-1β upregulation. However, the RNA probes we used would have detected both the CXCL12/SDF-1 isoforms described. The mechanism by which the suppression of CXCL12/SDF-1 by OSM occurs is not known. Clearly, this is a specific suppression since IL-6 was simultaneously induced strongly. Fibrocyte activation has also been shown through the engagement of CCR2 and its ligands CCL2/MCP-1 and CCL12/MCP-5 [46,47,48]. Others have shown in the FITC model of lung fibrosis that CCL12/MCP-5 was deemed to be the active ligand for CCR2 involvement in the fibrotic response in vivo [47]. Further analysis is required to assess the role of CCL12/MCP-5, possibly using inhibitors in vivo, to determine if it is central to the mechanism in the system with OSM overexpression.

## 5. Conclusions

The OSM-induced inflammation in BALB/c mouse lungs does not require IL-13; however, IL-13 is required for maximal extracellular matrix deposition. CD45+ coll1+ fibrocytes are induced by OSM but are also reduced in IL-13-/- mice. Chemotactic factors for fibrocytes other than CXCL12/SDF-1 may be involved since OSM selectively suppresses the CXCL12/SDF-1 expression in this system. These results suggest additional mechanisms in the OSM-induced ECM accumulation include the IL-13-dependent fibrocyte accumulation.

## Figures and Tables

**Figure 1 cells-08-00126-f001:**
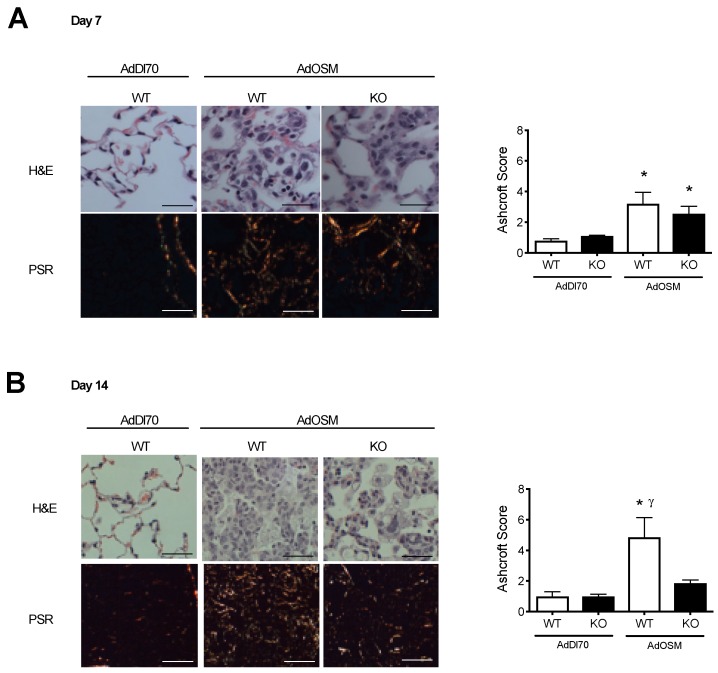
Prolonged induction of interstitial collagen in the lungs of AdOSM-infected BALB/c mice is reduced in IL-13-deficient mice. Wildtype (WT) and IL-13-/- (KO) BALB/c mice were treated with an endotracheal administration of AdDl70 or AdOSM (5 × 10^7^ pfu), culled at day 7 (**A**) or day 14 (**B**), and tissues were prepared for histology. Images of H&E and picrosirius red (PSR) stained tissue sections are shown from mice treated with AdDl70 or AdOSM. Quantification of picrosirius red staining was completed by Ashcroft scores to assess the degree of staining in the lung parenchyma. Data i shown as the mean +/− SEM (*n* = 5 per group). White bars are wildtype and black bars are IL-13-/- data. Scale bars for photomicrographs represent a length of 100um. * indicates significant difference of *p* < 0.05, as compared to AdDl70-treated mice. γ indicates significant difference of *p* < 0.05 as compared to AdOSM-treated IL-13-/- mice.

**Figure 2 cells-08-00126-f002:**
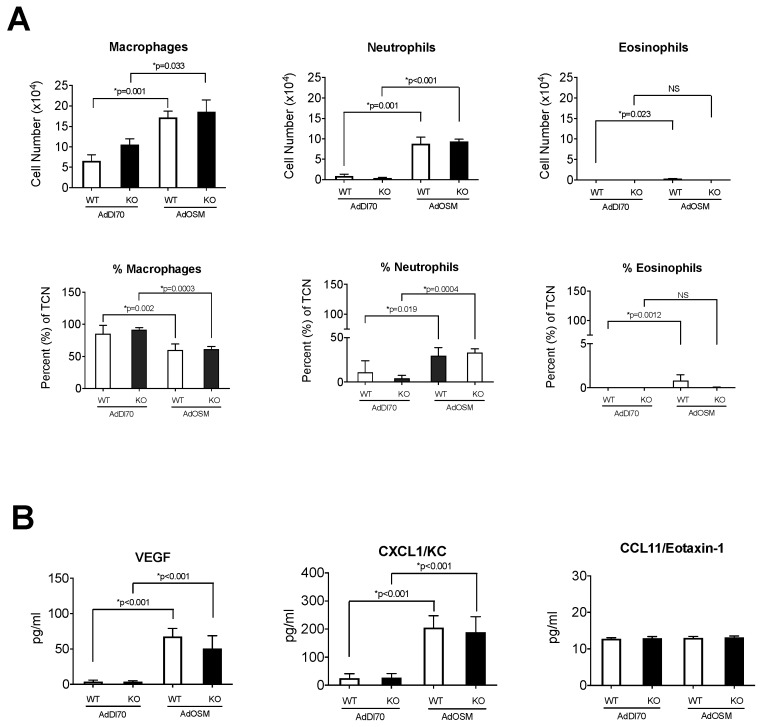
Inflammatory cells and cytokines in BALF at Day 7 in Response to AdOSM. (**A**) Wildtype (WT) and IL-13-/- (KO) BALB/c mice were treated with an endotracheal administration of AdDl70 or AdOSM (5 × 10^7^ pfu), culled at day 7, and bronchoalveolar lavage fluid (BALF) was collected and analyzed for cell numbers and percent of total cell number (TCN) of macrophages, neutrophils, and eosinophils in stained cytocentrifuge smears. (**B**) IL-6, CXCL1/KC and CCL11/Eotaxin-1 cytokine levels in BALF used in (**A**) was measured by ELISA. White bars are wildtype and black bars are IL-13-/- data. Data is shown as the mean +/− SEM (*n* = 5 per group). Statistical significant differences are noted with their p values between the indicated treatment groups.

**Figure 3 cells-08-00126-f003:**
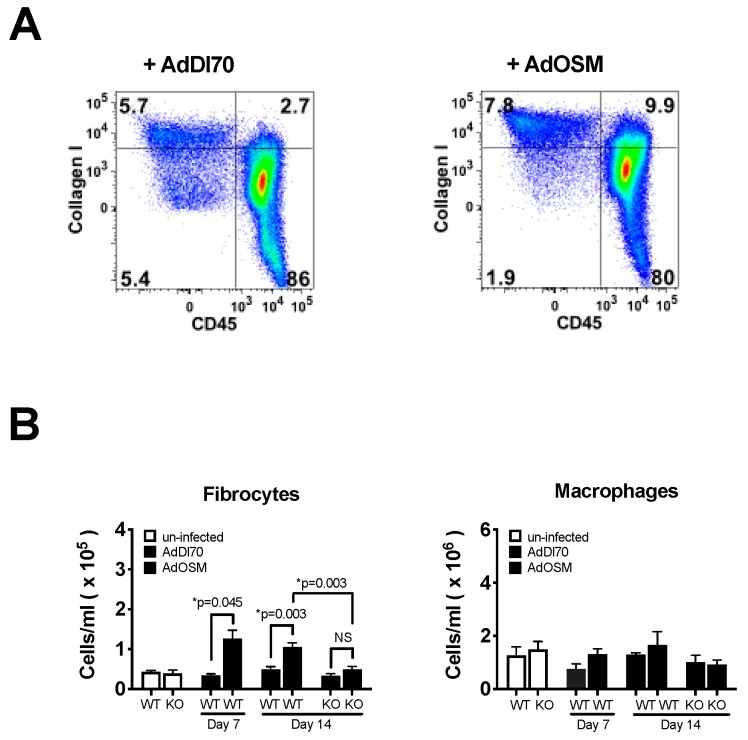
Fibrocyte numbers in AdOSM-treated BALB/c Mouse Lungs. Wildtype BALB/c mice untreated naive mice or wildtype and IL-13-/- (KO) treated mice were endotracheal administered AdDl70 or AdOSM (5 × 10^7^ pfu), culled at day 7 or 14, and mononuclear cell suspensions were analyzed by flow cytometry (as described in methods). (**A**) Representative flow cytometry plots showing CD45+col1+ populations (fibrocytes) at day 7 (**B**) CD45+ Collagen 1A1+ Fibrocytes and CD11c- CD11b+ Gr-1- DX5- Macrophage cells from whole lung tissue, after 7 and 14 day treatment, were also quantitated by flow cytometry. Data is shown as the mean +/− SEM (*n* = 5 per group). Statistical significant differences are noted with their p values between the indicated treatment groups. No statistical significance was observed between macrophage numbers in 3B.

**Figure 4 cells-08-00126-f004:**
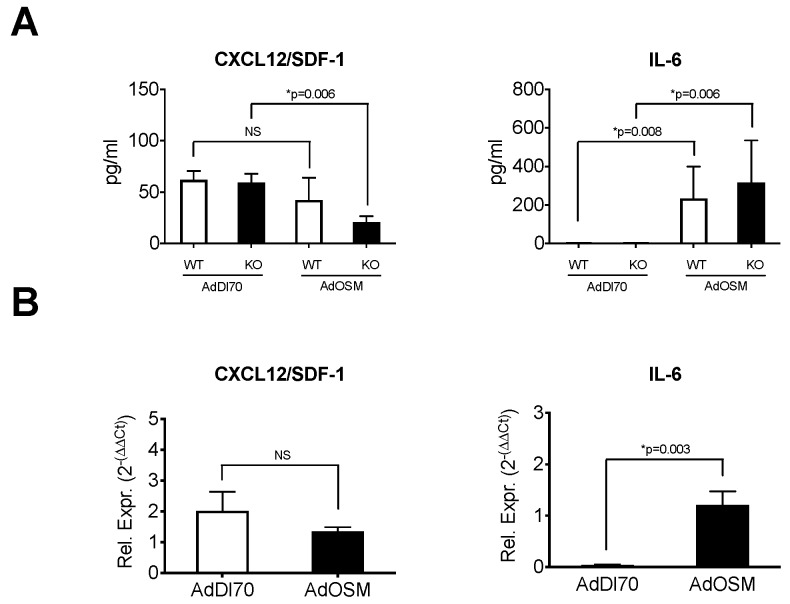
Regulation of fibrocyte chemokine mRNA by OSM. Wildtype BALB/c mice or IL-13-/- (KO) were treated with an endotracheal administration of AdDl70 or AdOSM (5 × 10^7^ pfu) as indicated, culled, BALF retreived and lung tissues were isolated and prepared for RNA extraction. (**A**) CXCL12/SDF-1 and IL-6 proteins in BALF were analyzed by ELISA. (**B**) RNA was probed for CXCL12/SDF-1 and IL-6 using q-RT-PCR. Rel. Expr.: Relative expression (Relative to 18S RNA). Data is shown as the mean +/− SEM (*n* = 5 per group. Statistical significant differences are noted with their p values between the indicated treatment groups.

**Figure 5 cells-08-00126-f005:**
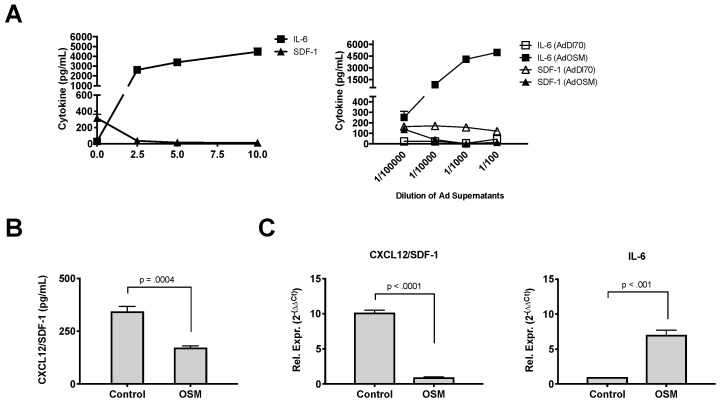
OSM downregulates expression of CXCL12/SDF-1 in mouse lung fibroblasts. BALB/c-derived mouse lung fibroblasts (MLF) were stimulated for 24 h (in triplicate) with increasing dosing of recombinant murine OSM (ng/mL) and supernatants analyzed by ELISA for CXCL12/SDF-1 or IL-6 (**A, left panel**). In (**A right panel**), MLF cells were incubated with dilutions of supernatants from AdDl70 (control)- or AdOSM-infected A549 cells (from 10^−5^ to 10^−2^), for 24 h, and MLF supernatants analyzed by ELISA for CXCL12/SDF-1 and IL-6. In (**B**), MLF cells derived from C57Bl/6 mice were stimulated for 24 h with recombinant OSM (10 ng/mL) and secreted CXCL12/SDF-1 protein was measured by ELISA. (**C**) CXCL12/SDF-1 and IL-6 mRNA expression in C57Bl/6-derived MLF cells was measured by quantitative PCR. Rel. Expr.: Relative expression (Relative to 18S RNA). Data represent one of at least three separate experiments. Data is shown as the mean +/− SD (*n* = 4 replicate treatments per group). Statistical significant differences are noted with their p values between the indicated treatment groups.

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
