# Peer review of "Extracellular Matrix and Fibrocyte Accumulation in BALB/c Mouse Lung upon Transient Overexpression of Oncostatin M"

_cells, 2019, doi:10.3390/cells8020126_

Round 1

Reviewer 1 Report

In their submitted manuscript, Botelho et al. investigate the pro-fibrotic activity of oncostatin M in Balb/c mice and its dependence on IL-13. They can clearly show that OSM (as already shown previously in C57Bl/6 mice) induces alveolar wall thickening, collagen deposition, neutrophil and macrophage infiltration as well as KC chemokine induction. At earlier time points (day 7) these effects are independent of IL-13, however, the sustained fibrotic response is alleviated in the absence of IL-13.

The manuscript is well written and provides not completely unexpected, but nonetheless important information. Some further points may be addressed to improve the manuscript.

Major points:

1)     The authors fail to provide data explaining a mechanism by which IL-13 might become important at later time points. In their discussion they mention that they failed to detect IL-13 in BALF and speculate that the local concentration might be too low or that IL-13 may be required at other organ sites. Two suggestions: 1) is it possible to detect Il13 mRNA levels in lung tissue or even better in isolated macrophages?

2)     With respect to the macrophages the authors show increased levels after adenoviral application of OSM in BALF (Fig. 2), however, the cell number does not change in the tissue (Fig. 3). Does the activation status of macrophages change upon OSM stimulation? Do the tissue resident macrophages express markers of M2 macrophages which could contribute to the fibrosis process?

3)     Fibrocyte accumulation (Fig. 3B) should be analyzed at day 7 also in IL13ko mice.

4)     Why are approx. 90% of single cells isolated from the lung tissue positive for CD45 (Fig. 3A)? Did the author pre-gate before on CD45-positive cells? What about stromal cells, epithelial cells? Additionally the axis is unclear in this figure. How can there be fluorescence below zero?

5)     A number of studies imply an induction of SDF1/CXCL12 gene expression in response to OSM. Could it be possible that the time point analyzed here is too late?

Minor points:

1)     Legend to figure 1: IL-33 instead of IL-13

2)     Figure 2A: if scales are anyway different for all plots then enlarge eosinophil plot

3)     Figure 3B: percentage of total cells is missing (depicted only cell numbers)

4)     Legend to figure 4: IL-1 instead of IL-6

5)     Scientific notation of 10exp5, 10exp6, 10exp7 has to be checked

6)     FACS marker for neutrophils are missing in material & methods

7)     The author might consider using CXCL1 instead of KC throughout their manuscript.

Author Response

Reviewer 1

Comments and Suggestions for Authors

In their submitted manuscript, Botelho et al. investigate the pro-fibrotic activity of oncostatin M in Balb/c mice and its dependence on IL-13. They can clearly show that OSM (as already shown previously in C57Bl/6 mice) induces alveolar wall thickening, collagen deposition, neutrophil and macrophage infiltration as well as KC chemokine induction. At earlier time points (day 7) these effects are independent of IL-13, however, the sustained fibrotic response is alleviated in the absence of IL-13.

The manuscript is well written and provides not completely unexpected, but nonetheless important information. Some further points may be addressed to improve the manuscript.

Major points:

1)     The authors fail to provide data explaining a mechanism by which IL-13 might become important at later time points. In their discussion they mention that they failed to detect IL-13 in BALF and speculate that the local concentration might be too low or that IL-13 may be required at other organ sites. Two suggestions: 1) is it possible to detect Il13 mRNA levels in lung tissue or even better in isolated macrophages?

Response:  As the reviewer and the manuscript note, the mechanism by which IL-13 is required in this model is not clear, and may involve IL-13 expression in specific populations (such as certain macrophage phenotypes) or SDF-1 (in certain cell populations) at specific time points, and have added the following to the discussion (line 279-281 in revised manuscript).

“It is possible that IL-13  (or SDF-1) is elevated at specific times not captured by the present time-points analysed, or indeed in specific populations (such as separate macrophage phenotypes).  Assessing specific macrophage populations or other cells over the time course of the model development would assist with this, and would be the subject of future experimentation.”

2)     With respect to the macrophages the authors show increased levels after adenoviral application of OSM in BALF (Fig. 2), however, the cell number does not change in the tissue (Fig. 3). Does the activation status of macrophages change upon OSM stimulation? Do the tissue resident macrophages express markers of M2 macrophages which could contribute to the fibrosis process?

Response: Although the cell numbers in the BAL  do increase in response to AdOSM ( to approximately 20,000 cells per lung, figure 2), the numbers are a relatively small part of the total lung tissue macrophage numbers (1-2 million cells per lung, Fig 3) which do not significantly change at Day 7.

The status of the tissue macrophages is an interesting point.  To the discussion, we have added the following in the revised manuscript line 250-256) 

“We have examined expression of macrophage markers (by mRNA analysis) in BALB/c mouse lungs treated with AdOSM and have not observed any significant change in alternatively activated “M2” markers (not shown). In contrast, C57Bl/6 mice show elevated CD206+ cells and arginase-1  expression (as typical M2 macrophage products) in response to AdOSM (reference#48) . Other macrophage phenotypes may be involved in BALB/c lungs in this system, however this would require further investigation.

3)     Fibrocyte accumulation (Fig. 3B) should be analyzed at day 7 also in IL13ko mice.

Response: While ECM accumulation was observed in day 7 AdOSM-treated mice, this was similar between wildtype and IL-13KO mice but reduced  in IL-13KO when compared to day-14 treated wildtype mice. Thus, while worth noting, we suggest that any differences in fibrocyte accumulation at early time points of AdOSM treatment leading to increases in ECM accumulation are IL-13 independent. A statement in the discussion has been added to address this (lines 273-277).

“Accumulation of fibrocytes over the time course of the ECM accumulation in this model may be affected at other time points in IL-13KO mice. The association with the decreased fibrocyte number and decreased ECM at day 14 does not show causality, and further experimentation to determine if fibrocytes are critical to the ECM accumulation at Day 14 would be required"

4)     Why are approx. 90% of single cells isolated from the lung tissue positive for CD45 (Fig. 3A)? Did the author pre-gate before on CD45-positive cells? What about stromal cells, epithelial cells? Additionally the axis is unclear in this figure. How can there be fluorescence below zero?

Response: Lung cell isolation for flow cytometry involved the use of 40um cell strainers. We have found that when using 40um cell strainers the lung cells isolated are primarily CD45+ hematopoietic cells. The use of a larger mesh size, such as 100um, indeed does increase the detected number of larger cells, such as epithelial and fibroblast stromal cells, but has in our hands effected the quality and running of samples through the flow cytometry. For clarity, we have added the following phrase  to the methods on line 108-109 in the revised manuscript:

….(this results in primarily CD45+ hematopoietic cells introduced to the flow cytometry analysis)”

In Fig 3A, the Flow analysis plots shown are in bi-exponential/logical format. This allows for detection of all fluorescence including “negative” (baseline) fluorescence that are a result of flow compensation calculations.

5)     A number of studies imply an induction of SDF1/CXCL12 gene expression in response to OSM. Could it be possible that the time point analyzed here is too late?

Please see response to point #1.

Minor points:

1)     Legend to figure 1: IL-33 instead of IL-13. 

Response: This has been corrected

2)     Figure 2A: if scales are anyway different for all plots then enlarge eosinophil plot.

Response: The scales have been adjusted for ease of comparison.

3)     Figure 3B: percentage of total cells is missing (depicted only cell numbers)

Response:  We have focussed on actual numbers of fibrocytes and macrophages here since we believe the load of these hematopoietic cells may be important in the lung in this model. 

4)     Legend to figure 4: IL-1 instead of IL-6.

Response: This has been corrected.

5)     Scientific notation of 10exp5, 10exp6, 10exp7 has to be checked.

Response: This has been checked throughout the manuscript

6)     FACS marker for neutrophils are missing in material & methods.

Response:  The FACS marker used for neutrophils, indicated in materials and methods, was Pacific Orange-conjugated anti-Gr-1. This has been indicated in the methods  (line 121)

7)     The author might consider using CXCL1 instead of KC throughout their manuscript.

Response: As suggested, this has been changed to CXCL1 throughout the manuscript and stated as CXCL1/KC within material and methods only.

Reviewer 2 Report

The Manuscript by Botelho et al. aims to highlight  the role of Oncostatin M in ECM accumulation in mouse lung. The manuscript is well written and easy to follow, the experimental plan is clear and elegant. I have just a few comments on the figures:

In figure 2A in the Eosinophils graph, the authors should split the y-axis in 2 so that the reader can appreciate the difference between the samples. I understand the willing of the authors to keep the same scale for all graph, but I think it should be improved.

Similarly in figure 2B all the y-axis scale should be rearranged following the data. For example in the Eotaxin-1 graph there are no values above 20 so I don't get why the scale go up to 50. Same problem for VEGF and KC graph.

Real time PCR methods is not properly described in methods section, in addition in figure 4B since none of the sample is equal to 1 I guess the author used a third sample as a control. Please clarify this point.

Author Response

Reviewer 2

Comments and Suggestions for Authors

The Manuscript by Botelho et al. aims to highlight  the role of Oncostatin M in ECM accumulation in mouse lung. The manuscript is well written and easy to follow, the experimental plan is clear and elegant. I have just a few comments on the figures:

In figure 2A in the Eosinophils graph, the authors should split the y-axis in 2 so that the reader can appreciate the difference between the samples. I understand the willing of the authors to keep the same scale for all graph, but I think it should be improved.

Response: As suggested, this has been revised: Figure 2A: Y-axis has been split for better comparison to neighbouring graphs. Figure 2A has also been updated to include % cells, as requested by another reviewer.

Similarly in figure 2B all the y-axis scale should be rearranged following the data. For example in the Eotaxin-1 graph there are no values above 20 so I don't get why the scale go up to 50. Same problem for VEGF and KC graph.

Response: This has been revised: Figure 2B: Y-axis for VEGF and KC (now labeled as CXCL1 through the manuscript, as requested by another reviewer) were enlarged in order to accommodate statistic comparisons within the graph. The Y-axis for Eotaxin-1 has been changed to match that of CXCL1 and to simply reflect that there is low levels and no significant change in Eotaxin-1.

Real time PCR methods is not properly described in methods section, in addition in figure 4B since none of the sample is equal to 1 I guess the author used a third sample as a control. Please clarify this point.

Response: The methods have been revised: A reference has been added to the Real time PCR methods that describes the method in detail. The relative mRNA expression is expressed as the DDCt for SDF-1 or IL-6 relative to that of 18S control RNA. A statement has been added to the methods sections to indicate this (lines 102-105).

Reviewer 3 Report

The concept of this manuscript is investigating the mechanisms of ECM accumulation in lung by transient over-expressed Oncostatin M (OSM).  Previously authors transfected the adeno-OSM to the BALB/c and C57Bl/6 mice and constructed the ECM accumulation mouse model eliciting by OSM over-expression. In this study, authors used IL-13 knockout mouse to study the effect of IL-13 in ECM accumulation by OSM over-expression. ECM accumulation was induced at Day7 in both WT and IL-13 knockout mouse. But at Day14, although ECM accumulation was continuously induced in WT mouse, it was decreased in IL-13 knockout mouse suggesting that IL-13 required to the maximal ECM accumulation. The both CD45 and collagen1 positive fibrocytes were reduced in IL-13 knockout mouse. Further inflammatory cytokine of IL-6 was increased but chemotactic of SDF-1 was decreased in adeno-OSM model.

These results are interesting, but reviewer has some questions about experimental design and several figures should be prepare more carefully.  Detail comments are below.

Major points

1, Figure3B right panel shows the Adeno-OSM induces the number of macrophages than control Adeno (AdDi70).  However, control Adeno (AdDi70) reduces the number of macrophages than non-adeno treated mouse.  And the macrophages on non-adeno and on Adeno-OSM looks almost same. This suggest that Adeno-OSM is involved in the recovery mechanisms by damage from adeno virus infection. Both IL-6 and SDF-1 (Fig 4 and 5) may be affected by macrophages. Authors should test non-infected WT and KO mice to check the side effect of adeno virus in Fig1, 2, 3A, 4, and 5.

2, At Day 7, ECM accumulation in Adeno-OSM treated IL-13 knockout mouse was same level with Adeno-OSM treated WT mouse in Fig1. At Day 14, ECM accumulation in Adeno-OSM treated IL-13 knockout mouse was decrease, but it was significantly increased in Adeno-OSM treated WT mouse. From these, comparing the fibrocytes and macrophages between WT and KO mice at Day7 may be important. Authors should show the number of collagen1 positive cells in KO mouse at Day7 Fig3B left panel, and discuss about these points.

Minor points

1, Figure 1; The magnification of A and B were clearly different. Authors are better to change the pictures with same magnification and show the scale bar. Pictures of AsDI70 treated WT mice and non-treated mice should be showed.  There are “AdmOSM” and “AdOSM”. What is difference between these?

2, Figure 2; ECM accumulation at Day14 in WT was significantly increased comparing with Adeno-OSM treated mouse in Fig1. Inflammatory cells and cytokines at Day14 are better to test. Figure2B was not cited in the manuscript.

3, Figure 3; The cell numbers of KO mouse at Day7 should be include. The gap between WT and KO at Day 14 were different between right and left panels.

4, Figure 5; The dilution ration in Fig5A right panel should be listed. Manuscript describe the Figure 5B left panel, but there is one panel in Fig5B.

5, Authors should correct manuscript carefully. ex.

-“Day seven” and “Day 7”, “AdmOSM” “AdOSM” ”Ad-mOSM”, were mixed in the manuscript. Please check other words also.

-pp2 line14 “There is increasing evidence that conditions that involve ECM remodeling involve the ,,,,,,,”

Author Response

Reviewer 3

Comments and Suggestions for Authors

The concept of this manuscript is investigating the mechanisms of ECM accumulation in lung by transient over-expressed Oncostatin M (OSM).  Previously authors transfected the adeno-OSM to the BALB/c and C57Bl/6 mice and constructed the ECM accumulation mouse model eliciting by OSM over-expression. In this study, authors used IL-13 knockout mouse to study the effect of IL-13 in ECM accumulation by OSM over-expression. ECM accumulation was induced at Day7 in both WT and IL-13 knockout mouse. But at Day14, although ECM accumulation was continuously induced in WT mouse, it was decreased in IL-13 knockout mouse suggesting that IL-13 required to the maximal ECM accumulation. The both CD45 and collagen1 positive fibrocytes were reduced in IL-13 knockout mouse. Further inflammatory cytokine of IL-6 was increased but chemotactic of SDF-1 was decreased in adeno-OSM model.

These results are interesting, but reviewer has some questions about experimental design and several figures should be prepare more carefully.  Detail comments are below.

Major points

1, Figure3B right panel shows the Adeno-OSM induces the number of macrophages than control Adeno (AdDi70).  However, control Adeno (AdDi70) reduces the number of macrophages than non-adeno treated mouse.  And the macrophages on non-adeno and on Adeno-OSM looks almost same. This suggest that Adeno-OSM is involved in the recovery mechanisms by damage from adeno virus infection. Both IL-6 and SDF-1 (Fig 4 and 5) may be affected by macrophages. Authors should test non-infected WT and KO mice to check the side effect of adeno virus in Fig1, 2, 3A, 4, and 5.

Response: The numbers of macrophages detected in the flow cytometry were not significantly different between any of the groups shown (line 189-190). The apparent  decrease between naïve and AdDl70 groups was not statistically significant. A statement within the legend of Figure 3 has been added to express this (line 181)

2, At Day 7, ECM accumulation in Adeno-OSM treated IL-13 knockout mouse was same level with Adeno-OSM treated WT mouse in Fig1. At Day 14, ECM accumulation in Adeno-OSM treated IL-13 knockout mouse was decrease, but it was significantly increased in Adeno-OSM treated WT mouse. From these, comparing the fibrocytes and macrophages between WT and KO mice at Day7 may be important. Authors should show the number of collagen1 positive cells in KO mouse at Day7 Fig3B left panel, and discuss about these points.

 Please see response to Reviewer 1, major point #3.

Minor points

1, Figure 1; The magnification of A and B were clearly different. Authors are better to change the pictures with same magnification and show the scale bar. Pictures of AsDI70 treated WT mice and non-treated mice should be showed.  There are “AdmOSM” and “AdOSM”. What is difference between these?

Response: In the revised manuscript: instances of “AdmOSM” within the text have been corrected to “AdOSM” referring to  the Ad vector expressing mouse OSM. Magnification for 1A and 1B has now been corrected and scale bars have been added to the revised figure.

Although high doses of adenovirus vectors in general may themselves induce responses, we have consistently found in our previous work (references 23,24,38,48)  that the low viral titre that we use (10exp7 PFU) in infections with AdDl70 control virus do not show any statistical differences in cytokine levels or cell accumulation in BAL, nor any detectable histological differences from control naïve mice at day 7 or 14 in either C57Bl/6 or BALB/c mouse lungs.

2, Figure 2; ECM accumulation at Day14 in WT was significantly increased comparing with Adeno-OSM treated mouse in Fig1. Inflammatory cells and cytokines at Day14 are better to test. Figure2B was not cited in the manuscript.

Response:  We have examined the BAL at Day 14 after AdOSM and find no significant   changes in the percentage of macrophages, neutrophil, lymphocytes  or  eosinophils  compared to AdDel70  in BALB/c mice. Figure 2B is now cited within the text of the results section.

3, Figure 3; The cell numbers of KO mouse at Day7 should be include. The gap between WT and KO at Day 14 were different between right and left panels.

See response to Reviewer 1, major point #3. The Gap between WT and KO at Day 14  in Fig 3 is now revised to be comparable between right and left panels.

4, Figure 5; The dilution ration in Fig5A right panel should be listed. Manuscript describe the Figure 5B left panel, but there is one panel in Fig5B.

Response: In the revision, the dilution ratio has been added within the x-axis of Figure 5A right panel. The “left panel” referral to Fig 5B has been deleted in the text.

5, Authors should correct manuscript carefully. ex.

-“Day seven” and “Day 7”, “AdmOSM” “AdOSM” ”Ad-mOSM”, were mixed in the manuscript. Please check other words also.

Response: The text has been revised throughout the manuscript: “Day seven” replaced by “Day 7”. “AdmOSM” and “Ad-mOSM” replaced by “AdOSM”. “AdDel70” replaced by “AdDl70”.

-pp2 line14 “There is increasing evidence that conditions that  are involved in ECM remodeling involve the ,,,,,,,”

This has been reworded to read:  “There is increasing evidence that conditions that  result in ECM remodeling involve the ,,,,,,,”

Round 2

Reviewer 1 Report

The revised manuscript by Botelho et al. addresses most points requested by the reviewers. A few minor points remain to be addressed:

1)      The authors should consider using the common nomenclature for all chemokines (i.e. Eotaxin-1, KC, SDF-1 and IL-8). If the authors want to keep the old nomenclature it might be useful to depict them as: CCL11/Eotaxin-1, CXCL1/KC, CXCL12/SDF1, CXCL8/IL-8

2)    Initial point 2) With respect to the macrophages the authors show increased levels after adenoviral application of OSM in BALF (Fig. 2), however, the cell number does not change in the tissue (Fig. 3). Does the activation status of macrophages change upon OSM stimulation? Do the tissue resident macrophages express markers of M2 macrophages which could contribute to the fibrosis process? Response: The status of the tissue macrophages is an interesting point. To the discussion, we have added the following in the revised manuscript line 250-256). We have examined expression of macrophage markers (by mRNA analysis) in BALB/c mouse lungs treated with AdOSM and have not observed any significant change in alternatively activated “M2” markers (not shown).

Request: Since this is an important point, the authors should include this data into the manuscript.

3)      Initial point 6) FACS marker for neutrophils are missing in material & methods. Response: The FACS marker used for neutrophils, indicated in materials and methods, was Pacific Orange-conjugated anti-Gr-1. This has been indicated in the methods (line 121). Request: The request was to define the complete gating strategy for neutrophils as mentioned for macrophages (line 126)

4)      Figure 2A,B: Regarding my initial comment and the comment of reviewer 2, it does not make much sense to reduce the resolution of the y-axis for neutrophils/eosinophils/VEGF/CXCL1/Eotaxin-1. The idea of both reviewers was to increase the resolution on the y-axis for better comparison. Figure 2A, lower panel: color coding is wrong (compared to upper panel).

5)      Initial minor point 3) Figure 3B: percentage of total cells is missing (depicted only cell numbers).  Response: We have focussed on actual numbers of fibrocytes and macrophages here since we believe the load of these hematopoietic cells may be important in the lung in this model.  Request: then the authors need to delete in line 199 of the revised manuscript that they depict the percentage of total cells in the right panel of Fig. 3B.

Author Response

Response to reviewer points:

1)  We agree the nomenclature must be consistent and thus have included the new and old nomenclature in the revised manuscript text,  with the designations CXCL1/KC, CXCL12/SDF-1, CCL2/MCP-1, CCL12/MCP-5, CCL11/eotaxin-1, CCL24/eotaxin-2.

These modifications were also completed  in the Fig 2,4,5 graph labels as well as the relevant figure legends.

2) We have now assembled data for  Supplemental Figure 1  to address the question of M2 macrophage phenotypes. The Figure shows RNA levels for  Arginase 1 (M2 marker) in total BALB/c lung is not significantly/statistically different for AdOSM vs AdDl70  control (ANOVA). For comparison, the Arginase-1 mRNA in C57Bl/6 lungs was induced markedly (P<0.005) consistent with our previously published work (reference 48).  In addition, mRNA from alveolar macrophages retrieved from AdOSM treated BALB/c mice showed no change mRNA levels for Arginase-1 or CD206.  Thus,  the macrophage populations BALB/c mice did  show evidence of alternatively activated/M2 phenotypes in this system. 

Thus we have added the supplemental figure, and added the following text to the results (revised manuscript lines 200-207) .  

"We have previously observed  significant increases in alternatively activated/M2 macrophage markers in C57Bl/6 mice treated with AdOSM (48). However, as shown in Supplemental Figure 1,  Arginase-1 mRNA was not significantly elevated in AdOSM-treated BALB/c lungs, while C57Bl/6 mice showed marked increases (p<0.005) which is consistent with previous results (48). In addition, mRNA from alveolar macrophages retrieved from AdOSM-treated  BALB/c mice did not show any difference in Arginase-1 or CD206 mRNA (both markers of M2 macrophage phenotypes) from those alveolar macrophages retrieved from AdDl70 treated animals." 

And to the discussion (line 287-288 in revision)  we have  replaced "(not shown)" with " (see Supplementary figure 1) 

3)  Although we have not  included data specifically on neutrophils in the Flow Cytometry results (Fig 3), we have included the gating strategy in the  methods section as requested (line 127-128)

"Neutrophils were defined as CD45+, CD11b high and Gr-1+."

4)  We have revised figure 2A (top panels) and 2B  to show each graph with a reduced  y axis (lower  maximum)  to increase resolution  while maintaining statistical comparisons within the graph axes.

In Fig 2A lower panel, the colour codes are now corrected in the revised figure.

5)  Regarding Fig 3, as correctly pointed out, the phrase regarding percentages is incorrect, and thus the sentence  in the revised  manuscript (line 197)   has the referral  to percentages removed.